# Sprout Suppressants in Potato Storage: Conventional Options and Promising Essential Oils—A Review

**Jena Thoma * and Valtcho D. Zheljazkov** 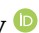

Department of Crop and Soil Science, Oregon State University, 109 Crop Science Building, 3050 SW Campus Way, Corvallis, OR 97331, USA; valtcho.jeliazkov@oregonstate.edu
* Correspondence: jena.thoma@oregonstate.edu

**Abstract:** Potatoes are a staple in the diet of millions, and constant demand necessitates the storage of large quantities to meet year-round consumption. Potato sprouting during storage is a major problem that leads to lost revenue and food waste, inspiring numerous studies into methods of sprout suppression. As bans on common synthetic suppressants become increasingly widespread, greater attention is turning to organic alternatives including essential oils (EOs) as sprout suppressants. This review presents an overview of physical and chemical means of sprout suppression in stored potato and critically analyzes studies focusing on the use of EOs for sprout suppression. Promising EOs are identified and evaluated for use in fresh, processing, and seed potato storage. Challenges and limitations of EO use in potato sprout suppression are discussed as well as areas of future research.

**Keywords:** potato storage; essential oils; sprout suppression; organic agriculture; review





## 1. Introduction

The domestication of potato (*Solanum tuberosum* L.) occurred over 8000 years ago in the Andean region of South America [1]. The Columbian Exchange introduced the potato to Europe, from which it spread to Asia and North America. Potato is currently the fourth largest crop after maize, wheat, and rice, and is cultivated in over 100 countries [2]. Potato plays an important role in global food security, reducing world hunger, and contributes to a balanced human diet and increased human health [3,4]. Enjoyed across the world, potatoes are highly versatile in cooking; they can be baked, boiled, and mashed, made into chips, or dehydrated into flakes and made into flour [1].

Given so many individuals' dependence on potatoes to meet their daily caloric needs, potato storage is crucial to ensure adequate supplies for consumption and seed. However, potato sprouting during storage is a major problem that leads to lost revenue and increased amounts of food waste [4]. In 2019, the world produced over 370 million megagrams (Mg) of potatoes, up from 333 million Mg in 2010 [5]. However, up to 25% of these totals may be lost from post-harvest to distribution [6]. Potato sprouting during storage plays a significant role in these losses, particularly when potatoes are stored in warm and humid climates which is often the case in developing countries [6].

Once harvested, potatoes may first undergo a "pre-storage" or curing phase where they are stored at 95% humidity between 10 and 15 °C for two weeks [2]. This phase allows potatoes to heal their peels after potentially being damaged during harvest while also allowing the potatoes to dry. Following pre-storage, potatoes are stored at low temperatures in piles or crates for periods ranging from several weeks to many months during which additional chemical or physical methods of sprout suppression may be applied prior to dormancy break [7].

Immediately following harvest, potatoes are in a naturally dormant state and will not sprout. However, the length of this innate dormancy is highly cultivar-dependent, and even the longest periods of innate dormancy generally do not last as long as is required by

processors and overall markets [2]. New models are emerging to predict potato dormancy length and forecast sprouting to inform storage management decisions, but sprouting is still a significant challenge [8].

Generally, dormancy in plants is a biological state in which plant growth is decreased or suspended even if the environmental conditions are favorable [9–11]. The dormancy period in potato is characterized by the absence of visible sprout growth and is under environmental, physiological, and hormonal control [10,12].

Sprouting in potato begins at the end of the dormancy period or when this period is interrupted by exogenous factors [11]. Control of sprouting during storage is necessary in order to prevent reductions in tuber quality and formation of toxic alkaloids, thereby eliminating food waste [10,12,13]. Potato sprouting involves the buildup of chlorophyll beneath the peel, a process known as "greening" [14]. As chlorophyll presence in potatoes is associated with solanine accumulation, an alkaloid that can be toxic to humans, green potatoes are considered inedible and become food waste [1]. Even sprouting without greening is undesirable as sprouting is accompanied by higher respiration rates resulting in potatoes that are smaller and more wrinkled [15]. For this reason, abundant research from the 20th century and onwards has focused on physical and chemical methods of sprout suppression.

Storage temperature and humidity level are the most critical factors regulating dormancy break during potato storage [16]. Storage between 8–12 °C at 85–90% relative humidity is the most appropriate and popular method for maintaining processing potato quality during long-term storage (up to 9 months) [15]. However, these conditions cannot prolong the dormancy period indefinitely, and once the natural dormancy period of the tubers has ended, storage temperatures of 8–12 °C allow for sprouting and sprout elongation [15]. Cold temperature storage will also alter sugar content, increasing glucose concentrations which cause products to fry dark resulting in unacceptable potato product color and economic losses [17]. Moreover, facility owners may lack the capital to install, run, or maintain cooling systems [18]. Other novel means of physical sprout suppression involve microwave irradiation [19], gamma irradiation [20], ultraviolet irradiation [21], pressure treatments [22], and treatment with magnetic fields [23] to physically damage the sprout buds, although these methods may quickly become expensive [2]. Furthermore, gamma irradiation is not allowed in certified organic systems. Therefore, chemical means of sprout control provide an effective and less expensive approach and have become necessary in maintaining potato quality during storage, no matter the destination market. Indeed, chemical sprout suppressants have been the most widely used method for potato sprout suppression in the last five decades.

This review examines various methods of sprout suppression in potatoes, with an emphasis on the use of essential oils (EOs) for potato sprout suppression. First, an overview of common synthetic suppressants is provided, followed by a list of organic suppressants with emphasis on commercially available EO sprout suppressants. EOs are then identified and evaluated for potential use in fresh, processing, and seed potato sprouting manipulation. Limitations to the use of EOs and avenues for future research are also discussed.

## 2. Chemical Means of Sprout Suppression

### 2.1. Commercial Synthetics

There are numerous synthetic chemicals used to achieve potato sprout suppression, including post-harvest sprout inhibitors such as isopropyl N-(3-chlorophenyl) carbamate (chlorpropham or CIPC), 1,4-dimethylnaphthalene (1,4-DMN), 2,6-diisopropylnaphthalene (2,6-DIPN), and 3-decen-2-one, as well as maleic hydrazide, a preharvest sprout inhibitor (Figure 1a).

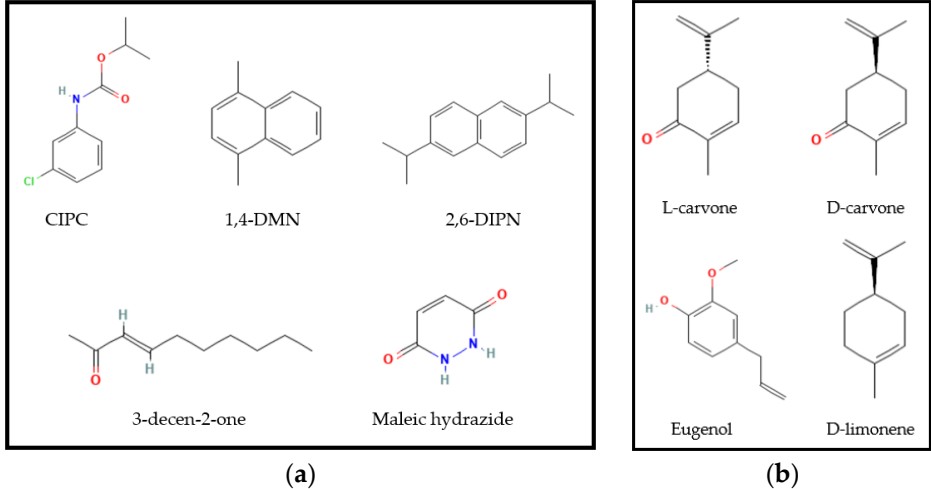

**Figure 1.** (**a**) Chemical structures representing the major synthetic chemicals used for potato sprout suppression during storage. (**b**) Chemical structures representing the major constituents of commercially available essential oil sprout suppressants.

### 2.1.1. Chlorpropham (CIPC)

CIPC is the most well-known and widely used sprout suppressant since the mid-20th century; 90% of all potato sprout suppressant applications in 2020 included CIPC in their mixtures [2]. Not only is CIPC inexpensive, but it is also highly effective due to its ability to interfere with cell division [24]. A single application of CIPC can completely inhibit sprout growth for up to 5 months due to its long-lasting residues, although a second application may be needed for storage periods longer than 6–8 months [12].

However, recent legislation in the USA and abroad has restricted the use of CIPC for sprout suppression. In the USA, the Food Quality Protection Act of 1996 lowered the amount of CIPC residue allowed on fresh potatoes from 50 ppm to 30 ppm [12]. While 30 ppm is still sufficient to ensure long-term sprout suppression, stricter proposals in the UK of 5 ppm might not be able to guarantee suppression longer than four months without repeat applications [12]. Moreover, the EU initiated an outright ban on CIPC, and final use of CIPC products was concluded in 2020 [25].

Stricter regulations come in response to research results demonstrating the harmful effects of CIPC residues and degradation products on human and environmental health [15,26,27]. Thermal degradation of CIPC during fogging application produces 3-chloroaniline (3-CA), a toxic compound that targets the hematopoietic and renal systems [26]. 3-CA is also produced by digestive activity when ingested by mammals, making CIPC residues potentially hazardous to humans [2]. Though there is no direct evidence of 3-CA as a carcinogen, its structural similarity to 4-CA, a known carcinogen and genotoxin, is cause for concern [2]. CIPC residues can be found on both fresh and processed potatoes alike. This includes finished products such as potato chips, as well as the cooking oil used to make French fries and the water used for washing [2]. Göckener et al. [27] demonstrated that CIPC could be reduced by 83%, 73%, and 78%, respectively, when potatoes were subjected to boiling, frying, and baking. Despite growing international concern and limitations on CIPC residues, CIPC still makes up a major portion of potato sprout control throughout the world. However, multiple synthetic alternatives exist.

### 2.1.2. Methyl-Substituted Naphthalenes

Potato tubers naturally produce methyl-substituted naphthalenes, such as 1,4-DMN and 2,6-DIPN that inhibit sprout activity through what is likely hormonal activity [12]. 1,4-DMN is a volatile compound that also contributes to the flavor and smell of baked potatoes [28]. Though these compounds occur naturally in potatoes in small amounts, synthetic formulations are generally used as plant growth regulator analogs to control

sprout growth in stored potatoes [2]. Unlike CIPC whose effects are considered permanent, and which only requires one or two applications, multiple applications of 1,4-DMN are necessary to inhibit sprout growth. In fact, multiple applications of higher-than-label rates were required to inhibit sprouting in potatoes stored above 4 °C [28]. Despite this, several studies demonstrate 1,4-DMN as an effective sprout suppressant [25,29]. In addition, the temporary nature of suppression renders 1,4-DMN a popular product for use in the seed potato industry, despite its potential to slightly delay plant emergence, reduce overall potato yields, and produce smaller potatoes [28]. While this effect could be desirable for some markets, the use of 1,4-DMN at higher-than-label rates is not legal in a commercial setting. For this reason, 1,4-DMN and 2,6-DIPN are often combined with CIPC to achieve lower CIPC residues while maintaining sprout suppression [12]. Despite the overall effectiveness of 1,4-DMN and 2,6-DIPN, little is known about the toxicity of their breakdown products. However, 1,4-DMN residues on treated potatoes may be low due to the compound's volatility [30].

### 2.1.3. 3-Decen-2-One

3-decen-2-one is a naturally occurring biopesticide found in certain mushroom species that is chemically synthesized for commercial sale [30]. Used as a food additive in processed foods, 3-decen-2-one is also an effective potato sprout suppressant that destroys the meristematic tissue of developing sprouts [30,31]. 3-decen-2-one is considered a safe and simple means of sprout suppression, and its adoption is growing in industrialized nations [15,31].

### 2.1.4. Maleic Hydrazide

Maleic hydrazide is another synthetic compound used for potato sprout suppression, both on its own or in combination with other treatments. Its effectiveness varies with cultivar and application timing, with some cultivars requiring multiple treatments to maintain sprout suppression [25,29]. This is likely the result of differences in innate dormancy among cultivars, with maleic hydrazide being particularly effective, even alone, for longer-dormant cultivars, and in combination with other suppressants for shorter-dormant cultivars [29]. Because maleic hydrazide is non-volatile, it can provide residual sprout control during shelf life of fresh potatoes even after the effects of volatile treatments such as 1,4-DMN have ended [29]. Indeed, treatment with maleic hydrazide showed significant residual suppression at nine months in all potato cultivars tested, and all other treatments, including CIPC, were more effective when combined with maleic hydrazide [29].

However, maleic hydrazide treatment can alter reducing sugar content. Some research suggests that maleic hydrazide treatment can increase reducing sugar concentrations in some cultivars [32], while other studies reported lowered reducing sugar content and better fry color as a result [33,34]. There are also reports of no differences in reducing sugar concentrations due to maleic hydrazide treatment, suggesting that its effects are likely dependent on cultivar or other factors [35,36]. Despite growing interest in its use, hydrazine, a derivative of maleic hydrazide, is a known carcinogen and mutagen, which may one day result in limitations on its use [2].

Maleic and l-tartaric acids were recently reported to have potential as potato sprout inhibitors. Immersion of potato tubers in solutions containing either acid for 18 h in the dark achieved sprout suppression for 6- and 4-weeks post-treatment at room temperature for maleic and l-tartaric acids, respectively [37]. Maleic acid is a derivative of maleic hydrazide oxidation and could underlie the effectiveness of the latter as a growth regulator, whereas the structural similarity of l-tartaric acid to maleic acid could explain its observed effectiveness [37]. However, the method utilized by the authors that involves 18 h soaking in water solutions would need to be examined thoroughly.

Despite the dominance of CIPC over the better half of a century, new laws and regulations aiming to curtail its use have spurred exploration into alternative means of sprout suppression. Various synthetic alternatives exist although none are quite as effective, and

all are more expensive [2]. Furthermore, the negative consequences that these alternative chemicals could have on human and environmental health have yet to be elucidated. As a result, a growing body of literature examining the use of organic alternatives for sprout suppression is emerging, and several products are already commercially available.

### 2.2. Commercially Available Sprout Suppressants for Certified Organic Systems

Certified organic production is a system that improves the health of soils, ecosystems, and people, and prohibits the use of synthetic pesticides and fertilizers, genetically modified organisms, artificial additives, and irradiation [38]. Organic sales reached $62 billion in the USA alone in 2021, an increase of 12.4% relative to 2020 [39]. While a limited number of synthetic products are allowed in organic production and processing, the previously mentioned synthetic potato sprout suppressants and irradiation are not allowed in organic operations. Currently, organic potato production utilizes hydrogen peroxide plus (HPP), ethylene gas, and various EO-containing products including Biox-M, Biox-C, Talent®, and ARGOS® [2], but may also employ edible coating technology in the near future [31,40].

### 2.2.1. Hydrogen Peroxide Plus (HPP)

HPP works by physically damaging the meristematic tissue in developing sprouts, thereby achieving temporary sprout suppression [2]. Like all organic methods of sprout control, HPP requires repeated or continuous applications to achieve adequate levels of suppression [12]. HPP breaks down into oxygen and water, making this product safe compared to the previously mentioned synthetic suppressants and their breakdown products.

### 2.2.2. Ethylene Gas

Studies on ethylene gas suggest that it can both enhance and suppress sprout growth [12]. These contradictory results are likely due to differences in cultivar, concentration, or timing of treatment. Rylski et al. [41] suggest that short-term treatment could reduce the length of dormancy, whereas continuous treatment achieved sprout suppression. Ethylene accounted for 15% of treatments to stored potatoes in 2016, with CIPC and spearmint (*Mentha spicata* L.) EO accounting for 82% and 3%, respectively [25]. However, the use of ethylene in processing potato storage may be limited as it may cause a slight darkening in frying color, reducing aesthetic appeal and desirability [30,42]. This effect may be inhibited by combining continuous ethylene with 1-methylcycloprene (1-MCP) application [30].

### 2.2.3. Biox-M

Of the trademarked EO products currently used for potato sprout suppression, Biox-M is the most common in the USA [25]. Biox-M contains 100% spearmint EO (*Mentha spicata* L.), whose active ingredient is R-(-)-carvone (L-carvone). Figure 1b displays the chemical structures of the major components in commercially available EO-containing sprout suppressants. Like HPP and ethylene gas, repeated or continuous treatment of Biox-M is required to maintain adequate suppression, and the high volatility of spearmint EO can hamper suppression efforts and increase costs [2]. Spearmint EO may also pose environmental risks if it is unknowingly released into waterways by plants washing treated potatoes; spearmint EO can have anesthetic effects on various fish species, presenting useful applications in aquaculture [43]. However, high doses of spearmint EO can cause stress in fish, raising cortisol levels, and potentially lead to death [44,45].

### 2.2.4. Biox-C

Biox-C is a commercial product containing 100% clove EO (*Syzygium aromaticum* L.), the main and active ingredient of which is eugenol. Repeated applications of Biox-C can achieve temporary suppression of sprouting with similar results as Biox-M [12]. The breakdown products of eugenol include ferulic acid, vanillin, and vanillic acid, which are considered safe [2]. However, like Biox-M, Biox-C could be harmful to aquatic life. Though

clove EO is commonly used as an anesthetic in aquaculture settings, if it is used improperly, it can cause fish mortality [46].

### 2.2.5. Talent®

Talent® is used in the Netherlands and Switzerland for sprout suppression and contains 100% caraway EO (*Carum carvi* L.) [47]. The active ingredient of caraway EO is S-(+)-carvone (D-carvone), the enantiomer of L-carvone in spearmint EO, and works similarly by damaging sprout meristematic tissue [12]. D-carvone breaks down into dihydrocarvone and dihydrocarveol, both of which are considered safe [2]. Furthermore, repeated applications of D-carvone can achieve sprout suppression for up to 274 days with similar efficacy as CIPC [48,49]. However, this method of sprout control is considerably more expensive than conventional CIPC treatment [50]. Nevertheless, the broader use of Talent® in organic potato storage is worth consideration.

### 2.2.6. ARGOS®

ARGOS® is the most recent commercial EO product for potato sprout suppression to enter the market, gaining approval for use in both the Netherlands and Ireland in 2020 [2]. ARGOS® contains orange EO (*Citrus* × *sinensis* L.), found in orange peels, of which the active ingredient is D-limonene. Orange EO is approved for use as a sprout suppressant in the USA and shows efficacy in longer dormant cultivars [29]. Orange EO is reported to show broad pesticidal activity, and concentrations of up to 10% are harmless to human health [51].

### 2.2.7. Aloe Vera Gel

Edible coatings containing Aloe vera gel were recently proposed as widely available and relatively inexpensive potential sprout suppressants for stored potato [31]. Edible coatings are easily applied to fruits and vegetables to prolong storage and shelf life by restricting gas exchange and limiting water loss [52]. Emragi et al. [40] observed favorable effects including sprout inhibition when potatoes were treated with edible coatings and stored at room temperature. Though this study did not investigate edible coatings containing Aloe vera gel, it is a promising new technology with strong potential for use in potato storage.

## 3. Essential Oils (EOs) for Sprout Control

The use of EOs in organic potato sprout suppression is not new, and several commercial products are available worldwide. However, because potato storage conditions vary depending on their end market use and the effectiveness of EOs depends on specific temperature and application schemes, their use is often unable to be generalized across all potato storage conditions [31]. What follows is a more thorough evaluation of studies focusing on EO efficacy for all types of stored potatoes, including the fresh, processing, and seed potato markets.

### 3.1. Essential Oils (EOs) for Sprout Control in Fresh Potatoes

To delay or minimize sprouting, potatoes destined for the fresh market are generally stored in the dark at temperatures between 3–7 °C and 85–90% relative humidity [18]. While these temperatures are effective in inhibiting sprout growth, they are not sufficient on their own to prevent sprout elongation, especially if potatoes are stored for 6 months or longer. Moreover, sprouting will occur once potatoes are placed in warmer environments such as in stores or consumer pantries [53]. For this reason, sprout suppressants are needed to ensure sprouting inhibition during long term storage and beyond. Table 1 summarizes the impact that various EO application regimens have on potato sprouting during storage. Spearmint (*Mentha spicata* L.), orange (*Citrus* × *sinensis* (L.) Osbeck), caraway (*Carum carvi* L.), and clove (*Syzygium aromaticum* (L.) Merr. and L.M. Perry) EOs have all been studied for their effectiveness in suppressing sprout growth in fresh potato storage.

**Table 1.** The impact of essential oil applications on potato sprouting, yield, and flavor.

| Essential Oil | Temperature (°C) | Combined With | Concentration | Potato Cultivars | Impact on Sprouting | References |
|---|---|---|---|---|---|---|
| Garlic (*Allium sativum*) | 20 | | 0.2 mg/mL | 'Favourita' | Reduced tuber sprout growth, and downregulated production of a protein involved in seed germination. | [54] |
| | 21–23 | | 2 mL/L | 'Gudene', 'Jalene' | Reduced sprouting in 'Jalene'. | [55] |
| Dill Seed (*Anethum graveolens*) | 5, 10, 15 | | Not specified | 'Agria' | More effective than spearmint EO although less effective than caraway EO at suppressing sprouting at higher storage temperatures with an effect similar to that of CIPC. | [49] |
| | 10 | | 25 mL/kg containing 4% dill seed EO, repeated after 5 weeks | 'Norland', 'Snowden' | Dill seed EO completely inhibited sprouting for 15 weeks in both cultivars with similar efficacy to CIPC and maleic hydrazide. | [56] |
| Dill Weed (*Anethum graveolens*) | 20 | | 32.5 mg/L airspace | 'Russet Burbank', 'Piccolo' | Reduced sprout growth by 50% over 29 days with effect similar to that of spearmint EO. | [57] |
| Caraway (*Carum carvi*) | 5–7 | | 100 mL/1000 kg every 6 weeks | 'Bintje' | D-carvone derived from caraway EO suppresses sprout growth just as effectively as CIPC for up to 274 days. | [48] |
| | 10 | | 25 mL/kg containing 4% D-carvone, repeated after 5 weeks | 'Norland', 'Snowden' | D-carvone completely inhibited sprouting for 15 weeks in both cultivars with similar efficacy to CIPC and maleic hydrazide. | [56] |
| | 10 | | 25 mL/kg containing 4% caraway EO, repeated after 5 weeks | 'Norland', 'Snowden' | Caraway EO completely inhibited sprouting for 15 weeks in both cultivars with similar efficacy to CIPC and maleic hydrazide. | [56] |
| | 5, 10, 15 | | Not specified | 'Agria' | Caraway EO was more effective than spearmint EO, dill EO, and CIPC at suppressing sprout growth at all three temperatures and could inhibit sprout growth for up to 180 days. | [49] |
| | 25 | | Not specified | 'Agria', 'Kennebec' | Moderate inhibitory effect on sprouting compared to untreated tubers, but not as strong as peppermint or coriander EOs. | [47] |
| *Chenopodium ambrosioides* | 24 | | 0.7 mL/L airspace | Not specified | Suppressed sprout growth for up to 10 weeks. | [58] |
| | 27 | | 0.7 g/L airspace | 'Russet Burbank' | Suppressed sprout growth for up to 28 days. | [59] |

**Table 1.** *Cont.*

| Essential Oil | Temperature (°C) | Combined With | Concentration | Potato Cultivars | Impact on Sprouting | References |
|---|---|---|---|---|---|---|
| Orange (*Citrus sinensis*) | 4.5 | | 100 mL/ton (total 900 mL over 9 months) | 'Maris Piper', 'King Edward', 'Melody', 'Nectar' | Could achieve similar levels of sprout suppression as continuous ethylene, but only if combined with maleic hydrazide. | [29] |
| | 8 | | 100 mL/ton every 3 weeks for 5 months | 'Agria', 'Verdi', 'Innovator' | ARGOS (orange EO) provides good control of sprouting for up to five months compared to a control, although CIPC was more effective. | [30] |
| Coriander (*Coriandrum sativum*) | 12 | | 0.5 µL/L airspace | 'Agria' | Stimulated sprouting of tubers. | [60] |
| | 12 | | 2 µL/L airspace, every four weeks | 'Agria' | Sprout suppression for up to 3 months, with a weaker effect than CIPC treatment. | [60] |
| | 25 | | 230 mL/L of vapor | 'Agria', 'Kennebec' | Sprout suppression between 65–95% with an effect significantly stronger than that of caraway EO. | [47] |
| *Cymbopogon citratus* | 27 | | 0.7 g/L airspace | 'Russet Burbank' | Suppressed sprout growth for up to 28 days. | [59] |
| Palmarosa (*Cymbopogon martinii*) | 21–23 | | 2 mL/L | 'Gudene', 'Jalene' | Reduced sprouting in 'Gudene'. | [55] |
| | 25 | | 200 µL/desiccator | 'Chipsona' | Seven-day treatment could inhibit sprouting up to 14 days after treatment. | [18] |
| Citronella (*Cymbopogon nardus*) | 10 | | 30 µL/L airspace, repeated after 35 days | 'Atlantic' | Completely suppressed sprout growth for up to 30 days after dormancy break. | [61] |
| Lemongrass (*Cymbopogon schoenanthus*) | 25 | | 200 µL/desiccator | 'Chipsona' | Enhanced sprouting with nearly 100% of eye germination a full day earlier than the control. | [18] |
| *Lippia multiflora* | 24 | | 0.7 mL/L airspace | Not specified | Suppressed sprout growth for up to 10 weeks. | [58] |
| | 27 | | 0.7 g/L airspace | 'Russet Burbank' | Suppressed sprout growth for up to 28 days. | [59] |
| Peppermint (*Mentha piperita*) | Not specified | | One pound EO per five tons potato per month | 'Russet Burbank' | Equally effective as spearmint EO (although less effective than CIPC) at sprout suppression. | [53] |
| | 8 | | 100 mg/kg | 'Asterix' | Menthol reduced sprout length and number for up to 50 days in non-dormant tubers. | [62] |
| | 10 | | 50 ppm/kg every two weeks | 'Asante', 'Kenya Mypa', 'Shangi' | Suppressed sprouting for 6 or 8 weeks compared to a control depending on cultivar. | [63] |

**Table 1.** *Cont.*

| Essential Oil | Temperature (°C) | Combined With | Concentration | Potato Cultivars | Impact on Sprouting | References |
|---|---|---|---|---|---|---|
| | 23 | | 50 ppm/kg every two weeks | 'Asante', 'Kenya Mypa', 'Shangi' | Suppressed sprouting for 8 weeks compared to a control, but sprouts were longer than those on potatoes stored at 10 °C. | [63] |
| | 25 | | 155 mL/L of vapor | 'Agria', 'Kennebec' | Sprout suppression between 65–95% with an effect significantly stronger than that of caraway EO. | [47] |
| Spearmint (*Mentha spicata*) | Not specified | | One pound EO per five tons potato per month | 'Russet Burbank' | Less effective than CIPC treatment, single application can enhance sprouting. | [53] |
| | 4.5 | | 60 mL/ton (total 360 mL over 9 months) | 'Maris Piper', 'King Edward', 'Melody', 'Nectar' | Controlled sprouting for all potato cultivars tested, performed equally well as 1,4-DMN and equally if better than CIPC. | [29] |
| | 4.5 | Ethylene | Ethylene 10 ppm, spearmint EO 60 mL/ton (total 180 mL over 9 months) | 'Maris Piper', 'King Edward', 'Melody', 'Nectar' | Acceptable suppression only possible when combined with maleic hydrazide application. | [29] |
| | 8 | | Initial application of 90 mL/ton followed by 30 mL/ton every three weeks or 45 mL/ton every four weeks (360 mL/ton total) | 'Agria', 'Verdi', 'Innovator' | Biox-M provides good control of sprouting for up to five months compared to a control, although CIPC was more effective. | [30] |
| | 8 | | Not specified | 'Bellini', 'Mondial', 'Désirée', 'Karlena', 'Eos', 'Nicola', 'Rodeo', 'Winston' | Monthly applications sufficient to inhibit sprouting in all cultivars tested over six months without significant reductions in salability although very low doses promote earlier axial sprouting, spearmint EO can be washed off with water to nullify its effects. | [64] |
| | 7, 9 | Ethylene | Spearmint EO 60 mL/ton (total 360 mL over six months), continuous ethylene | 'Innovator', 'Maris Piper', 'Performer', 'Royal', 'VR808' | Combination achieved better sprout control than either spearmint EO or ethylene alone but effectiveness depends on cultivar, just as effective or more effective at 7 °C than at 9 °C. | [25] |
| | 5, 10, 15 | | Not specified | 'Agria' | Efficacy of spearmint EO on sprouting decreases with increasing storage temperatures. | [49] |
| | 20 | | 21.5 mg/L airspace | 'Russet Burbank', 'Piccolo' | 50% reduction in sprout growth over a 29-day period. | [57] |

**Table 1.** *Cont.*

| Essential Oil | Temperature (°C) | Combined With | Concentration | Potato Cultivars | Impact on Sprouting | References |
|---|---|---|---|---|---|---|
| Black Spruce (*Picea mariana*) | Not specified | | 25% (*w/w*) | 'Colomba' | Suppresses sprout growth just as effectively as CIPC over 4 weeks when potatoes are stored at room temperature. | [65] |
| Rosemary (*Rosmarinus officinalis*) | 21–23 | | 2 mL/L | 'Gudene', 'Jalene' | Reduced sprouting in 'Jalene'. | [55] |
| Clove (*Syzygium aromaticum*) | Not specified | | 0.52 lb/5 tons potato | 'Russet Burbank' | Could achieve significant sprout control compared to untreated potatoes for up to 24 weeks. | [53] |
| | Not specified | | Initial application at 90 ppm followed by 30 ppm application three weeks later | Not specified | Achieved acceptable sprout suppression for 60 days. | [12] |
| | 8 | | 100 mg/kg | 'Asterix' | Eugenol reduced sprout length and number for up to 50 days in non-dormant tubers. | [62] |
| | 10 | | 120 or 240 mg/L airspace | 'Russet Burbank', 'Piccolo' | Biox-C application caused high levels of sprouting in the first week followed by bud necrosis and sprout suppression for up to 19 weeks. | [57] |
| | 25 | | 200 µL/desiccator | 'Chipsona' | Enhanced sprouting with nearly 100% of eye germination a full day earlier than the control. | [18] |
| *Thymus schimperi* | 21–23 | | 2 mL/L | 'Gudene', 'Jalene' | Reduced sprouting in 'Gudene'. | [55] |
| Ajwain (*Trachyspernun ammi*) | 25 | | 200 µL/desiccator | 'Chipsona' | Seven-day treatment could inhibit sprouting up to 30 days after treatment. | [18] |
| *Zingiber officinale* | 27 | | 0.7 g/L airspace | 'Russet Burbank' | Suppressed sprout growth for up to 28 days, with a stronger effect than *C. ambrosioides, L. multiflora*, or *C. citratus*. | [59] |

### 3.1.1. Spearmint Essential Oil

A study by Saunders and Harper [29] demonstrated spearmint EO as an effective sprout suppressant at 4.5 °C. Repeated applications of spearmint EO were able to adequately suppress sprouting over the 9-month storage period, and this strategy performed similarly to 1,4-DMN treatment despite needing more applications at higher concentrations. Spearmint EO treatment worked particularly well in longer dormant cultivars and produced acceptable suppression in shorter dormant cultivars, while comparable if not more effective sprout control than CIPC was achieved in all cultivars. However, the study only used a single application of CIPC.

Saunders and Harper [29] also applied spearmint EO in combination with continuous ethylene gas, utilizing half as many spearmint applications over the course of 9 months compared to spearmint EO on its own. Results indicate that the combination could suppress sprouting relative to a control, but acceptable suppression was only achieved if this treatment was combined with maleic hydrazide. The effect of the combination was stronger

than when ethylene was applied alone, but weaker than when spearmint EO was applied alone. The latter is likely due to the lower amount of total spearmint EO applied as a result of fewer total applications. Additional tests holding the total application of spearmint EO constant across treatments could better elucidate any additive effect of ethylene when used in combination with spearmint EO.

### 3.1.2. Orange Essential Oil

Saunders and Harper [29] also evaluated orange EO for sprout suppression at 4.5 °C. Results indicate that orange EO application could suppress sprouting to a similar level as continuous ethylene application, that is, some control in longer dormant cultivars. However, like the combination of ethylene and spearmint EO, orange EO could only achieve acceptable rates of suppression in these longer dormant cultivars when used in combination with maleic hydrazide.

### 3.1.3. Caraway Essential Oil

Caraway EO could be an effective sprout suppressant in fresh potatoes. Hartmans et al. [48] compared the effect of CIPC to D-carvone on potatoes stored at 5–7 °C. Results demonstrated that repeated applications of D-carvone can suppress sprout growth just as effectively as CIPC for up to 274 days. Carvone residues were highest in the peels and were directly related to the concentration applied; however, when these potatoes were peeled and cooked, no "off-flavor" was observed. While the study did not look at any off-flavors that might have been present in unpeeled, cooked potatoes, the D-carvone residue was significantly lower than that of CIPC, suggesting that the content was already very low. Caraway EO contains 50–60% D-carvone [47]. Therefore, it is possible that caraway EO could have similar effects on sprout suppression in fresh potato storage.

### 3.1.4. Clove Essential Oil

Clove EO may also be an effective sprout suppressant in fresh potato storage. Clove EO at an initial application rate of 90 ppm followed by an application of 30 ppm three weeks later can achieve sprout suppression for up to 60 days, with a performance similar to that of spearmint EO treatment [12]. However, this study did not indicate at what storage temperature these effects were observed.

There are several established organic methods of sprout suppression using EOs in fresh potatoes. EOs, when combined with the low storage temperatures often used for fresh market potatoes, can achieve effective sprout suppression in numerous potato cultivars. However, the higher temperature storage requirements for processing potatoes present challenges to the use of EOs in this market sector [49,63].

### *3.2. Essential Oils (EOs) for Sprout Control in Processing Potatoes*

Storing potatoes for processing markets poses unique challenges due to the need to maintain low levels of reducing sugars. Low storage temperatures can effectively delay sprout development, but also enhance hydrolysis of sucrose in the tuber flesh resulting in higher accumulations of reducing sugars that cause undesirable discoloration during frying [2]. Therefore, unlike fresh market potatoes, which can be stored at 3–7 °C, processing potatoes are stored between 8–13 °C to avoid tissue sweetening and subsequent revenue losses [50]. These higher temperatures necessitate chemical sprout suppressants, while also challenging their efficacy [49,63]. Though many of the EOs used for fresh potato storage are also used for processing potato storage, their efficacy often differs. In addition, EOs from peppermint (*Mentha x piperita* L.), dill (*Anethum graveolens* L.), palmarosa (*Cymbopogon martini* (Roxb.) Wats.), ajwain (*Trachyspermum ammi* (L.) Sprague ex Turrill), *Lippia multiflora*, *Chenopodium ambrosioides*, *Cymbopogon citratus*, *Zingiber officinale*, citronella (*Cymbopogon nardus* L.), rosemary (*Rosmarinus officinalis*), *Thymus schimperi*, garlic (*Allium sativum* L.), and black spruce (*Picea mariana* Mill.) show potential for use in this industry [18,55,59,61,65].

### 3.2.1. Spearmint Essential Oil

The effect of spearmint EO on sprouting at higher temperatures is uncertain. In one study, monthly applications of spearmint EO over six months at 8 °C were sufficient to inhibit sprouting in all eight cultivars tested, although to varying degrees [64]. The treated potatoes showed 38% less weight loss on average than untreated potatoes and remained firm enough to remain marketable through the end of the study. Visse-Mansiaux et al. [30] reported similar results; repeated applications of either Biox-M or ARGOS® suppressed sprouting for up to five months. However, Şanlı and Karadogan [49] suggest that the efficacy of spearmint EO decreases with increasing storage temperatures. This is in contrast with results suggesting a 50% reduction in sprout growth relative to a control over 29 days after a single treatment of spearmint EO despite the tubers being stored at 20 °C [57].

A combination of spearmint EO and continuous ethylene has also been evaluated at 7 °C and 9 °C [25]. Repeated applications of spearmint EO in combination with continuous ethylene achieved better sprout control over six months than either spearmint EO or continuous ethylene alone, although the effectiveness varied by cultivar [25]. In addition, the combination was either just as effective or more effective at 7 °C than at 9 °C, depending on the cultivar. These varying results suggest that spearmint EO, especially in combination with ethylene, could be a satisfactory means of sprout control for some cultivars destined for processing, although its efficacy at higher storage temperatures (>15 °C), alone or in combination with ethylene, is still uncertain.

### 3.2.2. Peppermint Essential Oil

Peppermint EO is a promising sprout suppressant for processing potatoes. A 155 mL/L vapor concentration of peppermint EO at 25 °C was associated with inhibition rates between 65% and 95% compared to the control [47]. Finger et al. [62] showed that treatment with menthol, a major constituent of peppermint EO [66], reduced sprout number and length for up to 50 days in non-dormant tubers. This is consistent with findings by Murigi et al. [63] reporting sprout suppression for up to 8 weeks after dormancy break at either 10 or 23 °C in tubers treated with peppermint EO.

Comparisons of peppermint and spearmint EO to CIPC also exist. At multiple application rates, peppermint EO was found to be equally effective as spearmint EO, while also resulting in fewer culinary and palatability issues [53]. While this study did not specify the storage temperature used, neither spearmint nor peppermint EO treatment resulted in negative sugar profile alterations or fry-color changes. Spearmint EO was found to negatively alter potato flavor compared to CIPC treatment, while peppermint EO caused no significant difference from CIPC [53]. Both peppermint and spearmint EO were less effective than CIPC treatment at suppressing sprout growth, but their effectiveness relative to the control encourages further investigation into their use, particularly peppermint EO, in processing potato storage.

### 3.2.3. Clove Essential Oil

Clove EO, too, is a promising sprout suppressant at the higher temperatures necessary for processing potato storage. Frazier et al. [53] reported that repeated applications of clove EO can achieve significant control of sprouting compared to untreated potatoes after 24 weeks but did not indicate at what storage temperature these effects were observed. Conversely, treatment with Biox-C and subsequent storage at 10 °C resulted in high levels of sprouting within the first week [57]. However, the emerging sprouts then decayed, resulting in decreasing numbers of sprouted tubers over the course of 29 days, and sprouting was controlled for up to 19 weeks [57]. This delayed effect is likely due to the lower volatility of eugenol, the active ingredient in Biox-C and a main constituent of clove EO, compared to L-carvone in spearmint EO. Finger et al. [62] found that eugenol application to non-dormant tubers reduced sprouting and sprout growth for up to 50 days at 8 °C. These studies point to the potential for clove EO as an effective sprout suppressant in processing potato storage.

### 3.2.4. Caraway Essential Oil

Caraway EO has also shown promise for sprout control at higher temperatures. Song et al. [56] found that a double application of either caraway EO or D-carvone could inhibit sprouting for 15 weeks at 10 °C, with similar efficacy to CIPC and maleic hydrazide treatments. In a study comparing caraway EO to spearmint and dill seed EOs at 5, 10, and 15 °C, caraway EO was the most effective at inhibiting sprout growth for up to 180 days at all three temperatures [49]. Caraway EO was also superior to CIPC, totally suppressing sprouting over the six-month study, even at 15 °C. In addition, caraway EO treatment was associated with the least amount of tuber weight loss, with increasing efficacy as temperatures increased. EO analysis in Şanlı and Karadogan's study [49] found carvone presence in all three EOs with the highest amount in spearmint EO and the lowest amount in caraway EO. Limonene was also present in all three EOs, with the highest amount in caraway, and the lowest amount in spearmint. Despite differing levels of carvone, EO applications in this study were adjusted to make the total amount of carvone applied the same among all treatments. The authors claim that the differences in the observed effects on sprouting are due to other compounds, such as limonene, acting on its own or in combination to produce sprout inhibition. However, the composition of EOs even within a species can vary widely and contain many compounds, some of which may be unique to only select cultivars [67,68]. As synergistic effects may exist among these various constituents, additional research should compare the effects of pure carvone and limonene to more fully understand how these compounds affect potatoes in storage. Another study reported moderate inhibitory effects of caraway EO at 25 °C, but that this effect was less than that of coriander or peppermint EO treatment [47]. This supports the notion that different EOs have specific temperature ranges in which they are most effective. Nevertheless, these results show the potential for caraway EO and EOs high in limonene, including dill seed and orange EO, for use in processing potato sprout suppression.

### 3.2.5. Dill Seed and Weed Essential Oils

Both dill (*Anethum graveolens*) seed and weed EOs have also shown potential to inhibit sprouting at higher storage temperatures. Repeated applications of dill seed EO can suppress sprouting for up to 15 weeks at 10 °C [56]. Şanlı and Karadogan [49] showed that dill seed EO was able to significantly suppress sprouting at 5, 10, and 15 °C, though to a lesser degree than caraway EO. However, its effect was similar to that of CIPC, which suggests it is a viable alternative. EO analysis in this study found dill seed EO to contain intermediate levels of both limonene and carvone, which seem to correspond to its intermediate performance over that of spearmint EO which has a high carvone content, and below that of caraway EO which has a high limonene content. Research on potatoes stored at 20 °C suggests that a single application of dill weed EO can reduce sprout growth by 50% over 29 days [57]. These studies show the consistent effectiveness of dill seed and weed EO as sprout suppressants at the higher temperatures required for processing potato storage.

### 3.2.6. Coriander Essential Oil

Goodarzi et al. [60] demonstrated coriander EO as an effective sprout suppressant at 12 °C. Repeated applications were able to suppress sprouting for up to three months, although this effect was weaker than that of CIPC [60]. Coriander EO has also shown notable effects on potato sprouting at 25 °C; a 230 mL/L vapor concentration of coriander EO achieved sprouting inhibition rates between 65% and 95% compared to the control [47]. The observed effects were significantly stronger than those of caraway EO treatment in this study, and no differences in taste or appearance were noted due to any of the treatments. However, this is a relatively high concentration, and similar sprout suppression at 25 °C may not be achieved at lower concentrations. Nevertheless, these results suggest that further research into coriander EO as a processing potato sprout suppressant is justified.

### 3.2.7. Emerging Essential Oils for Processing Potato Sprout Suppression

Owolabi et al. [58,59] identified *Lippia multiflora*, *Chenopodium ambrosioides*, *Cymbopogon citratus*, and *Zingiber officinale* as promising sprout suppressants at temperatures 24 °C and above. *L. multiflora* and *C. ambrosioides* were found to suppress sprouting for up to 10 weeks in one cultivar [58]. However, the suppressive effect of *Z. officinale* was the strongest of the four after 28 days, although none of the treatments completely suppressed sprouting [59]. Perhaps these effects would be stronger at lower storage temperatures or with repeated applications. For example, double application of citronella (*Cymbopogon nardus* L.) EO was able to completely suppress sprout growth for up to 30 days after dormancy break at 10 °C storage temperatures [61].

Some of the most recent EOs to show promise as sprout suppressants at 25 °C are palmarosa (*Cymbopogon martinii*) and ajwain (*Trachyspermum ammi*) [18]. A seven-day treatment with 200 μL of palmarosa EO in a 300 mm vacuum desiccator could inhibit sprout growth for 14 days after treatment whereas treatment with 200 μL ajwain EO for seven days could completely inhibit sprouting for up to 30 days after treatment [18]. Belay et al. [55] also reported sprout suppressive activity of palmarosa EO as well as that of *Thymus schimperi* and rosemary (*Rosmarinus officinalis*) EOs in select potato cultivars. These results suggest that repeated application of palmarosa and ajwain EOs could achieve longer term suppression.

Belay et al. [55] also reported reduced sprouting due to treatment with garlic (*Allium sativum*) EO. Results reported by Li et al. [54] corroborate these findings, but also suggest that garlic EO application causes downregulation of a gene shown to enhance seed germination when overexpressed in tobacco.

Black spruce EO (*Picea mariana*) may be another EO applicable for processing potato storage as it has been shown to completely suppress sprouting for up to four weeks at room temperature [65]. Further studies evaluating black spruce EO at specific temperatures or over a longer timeframe could more fully define its potential as a sprout suppressant.

Chemical sprout suppression is even more vital in processing potato storage than it is in fresh potato storage due to the need to store potatoes at higher temperatures. These higher temperatures present challenges to commonly used EO products, particularly spearmint EO, which shows inconsistency in its ability to suppress sprouting at temperatures above 15 °C. However, numerous EOs have shown exciting promise in sprout suppression at temperatures of up to 27 °C. This could allow for satisfactory storage of all potatoes no matter their market destination at ambient temperatures without the use of cold storage. This could save money on electricity, reducing the greenhouse gas emissions associated with current storage techniques, as well as expand the storage capabilities of small and marginal farmers and processors.

### 3.3. Essential Oils (EOs) for Sprout Control and Enhancement in Seed Potatoes

Unlike fresh or processing potatoes, seed potatoes must be stored and managed in a way that preserves their ability to sprout. If sprout suppressants are used, their effects must be reversible so that a crop may be produced in following seasons. Seed potatoes can be stored between 2–4 °C which prevents sprouting and sprout growth [15]. However, this may not be feasible due to cooling and infrastructure costs [18]. While the fact that multiple applications of EOs are necessary to maintain sprout suppression is often seen as a disadvantage to their use in both fresh and processing potato storage, this fact presents a unique opportunity for seed potato storage as sprouting can often resume on its own in a matter of weeks after EO applications cease. Studies in this realm often report effects on tuber yields, while a few also look at the potential for EOs to enhance sprouting.

### 3.3.1. Sprouting Enhancement

As spearmint EO is the most used EO in potato storage, several studies evaluated the effect of spearmint EO on sprout enhancement and subsequent tuber yield. Teper-Bamnolker et al. [64] found that the inhibitory effects of spearmint EO can be nullified

by washing treated potatoes with water, after which sprouting resumed within days although with reduced apical dominance. Interestingly, very low doses of spearmint EO may promote earlier axial sprouting [64], and even a single application of spearmint EO can enhance sprouting [53]. These results indicate the importance of repeated spearmint EO applications in prolonged sprout suppression, but also suggest their utility in seed potato sprout enhancement. Spearmint EO application is associated with a concentration-dependent delay in emergence with higher concentrations causing the longest delays with no significant effect on tuber yield [57].

Low concentrations of coriander EO may also enhance sprouting. Goodarzi et al. [60] found that coriander EO applications of 0.5 μL/L airspace stimulated tuber sprouting at 12 °C storage temperatures. These studies demonstrate spearmint and coriander EO's suitability as sprout suppressants until seed potato planting as well as their potential as sprout enhancers.

EOs could be combined with strategies utilizing red and far-red light-emitting diodes for suppression of sprout elongation in seed potatoes to achieve tight control over sprouting just prior to planting [69]. Whether this combination of strategies could be managed to increase yields in some cultivars remains to be seen.

### 3.3.2. Tuber Yield Enhancement

Sprouting and yield enhancement may be possible with West Indian lemongrass (*Cymbopogon schoenanthus* Spring.), and clove EO treatments. Shukla et al. [18] showed that lemongrass or clove EO treatment at 25 °C could enhance sprouting relative to a control, with nearly 100% of eye germination occurring a full day earlier than the control. Not only did both the lemongrass and clove EO treated potatoes exhibit more and longer sprouts than the control, these treated tubers also produced significantly higher potato yields, with lemongrass EO treatment resulting in slightly higher yields than clove EO treatment [18]. Quantitative real-time PCR analysis revealed that both clove and lemongrass EO treated potatoes up-regulated genes encoding for ethylene response factor (ERF), auxin-repressed protein (ARP), Aux/IAA proteins (AIP), and ADP-ribosylation factor (ARF), proteins associated with dormancy break in potatoes [18].

Recent research suggests that dill weed EO treatment of seed tubers can positively impact tuber yield [70], although this effect was not observed by Song et al. [57], perhaps due to differences in cultivar or application schemes. However, the potential for EOs to not only suppress sprout activity, but also to up-regulate it for the purpose of enhancing yields represents an exciting new direction of research.

Sprout suppression in seed potatoes is often necessary to maintain seed potato stores until the appropriate planting date. Recent studies have pointed to EOs as a means of achieving this as well as enhanced yields. This could lead to higher levels of production, a perpetual goal of agriculture, made more salient in light of a changing climate causing more variable yields.

### 4. Limitations of Essential Oils (EOs) and Areas of Future Research

Despite the potential for extensive use of EOs in potato sprout management, their widespread adoption is hampered by the diversity of EO compositions even within a single plant species, the lack of application schemes for various cultivars, and in some cases the high costs associated with their purchase and use. However, the need for standardization of EO composition within a species can be easily adjusted by the traditional EO companies that are usually handling EO between the producers and the end users.

While there is general similarity in the EO chemical profile within a species, the actual concentrations of various EO constituents within a single species can vary greatly, and may also be affected by climatic and soil conditions, season, stage of vegetative cycle, and production practices used [71]. Therefore, more research is needed to identify suitable species, a cultivar or chemotype within a species, and then tune up the production technology. Furthermore, some countries and regions are known for the production of

specific EOs and could provide consistency of supply and quality, e.g., peppermint and spearmint EO production in the Pacific Northwest USA or orange EO in Florida. While the mechanisms by which EOs are acting to achieve sprout suppression or enhancement are being studied, more research in this area is necessary to identify the EO constituents or group of constituents that are the most important and if synergistic interactions between the varying components exist.

The effectiveness of an EO treatment on sprout suppression likely depends largely upon its composition and is directly influenced by the concentrations of various compounds within the EO and possibly the ratio between key compounds. As these concentrations can vary greatly, this can lead to difficulties in determining universal application schemes even for a single species. Furthermore, EO efficacy can vary among potato cultivars, with a single EO showing adequate sprout suppression in one cultivar but showing inadequate suppression in another. This makes matching the right EO to an appropriate potato cultivar crucial. The correct concentration also needs to be determined to achieve sprout suppression while minimizing costs. Additionally, several studies used application schemes above legal limits to achieve sprout suppression. For example, spearmint EO applied at a rate of 60 mL/ton achieved adequate sprout suppression in various cultivars, but the maximum individual dose of the commercially available spearmint EO product, Biox-M, is 9 mL/ton in the UK [72]. This poses challenges to businesses, as ignoring the labeled rate is unlawful, although it is portrayed as necessary to ensure satisfactory suppression. It may be possible to combine various EOs or selected EO fractions into a single application to obtain synergistic effects while also remaining at or below the labeled rate for each. However, research into effective combinations, exact ratios, and application schedules will be necessary before this method can be safely implemented.

Finally, EOs can be more expensive than their conventional, synthetic alternatives. Due to the necessity of repeated or continuous applications, many businesses may be unwilling to purchase the quantities necessary to treat thousands of tons of potatoes. Furthermore, the application technology and methodology for various EOs may differ from the existing infrastructure mainly used to apply CIPC, which could entail even greater costs to make the shift [12]. Indeed, the best application method for various EOs will need to be determined, especially for ones that have recently begun to show promise. Despite the potential for new EO products to be developed in the coming years, whether they will be cost effective in commercial settings remains to be seen.

## 5. Conclusions

Potatoes are a major world commodity upon which much of the global population depends. Therefore, it is necessary that an adequate supply of both fresh and processing potatoes is always available and the food waste resulting from inadequate storage be minimized or eliminated. As the crop's natural dormancy period is not sufficient to maintain year-round supplies, physical and chemical means of sprout suppression are necessary to meet this goal. However, concern over widely used potato sprout suppressing chemicals, primarily CIPC, and their harmful effects on human and environmental health, has resulted in greater attention towards EOs as alternative sprout suppressants. Indeed, the availability of natural products for sprout suppression in potato has been a major impediment towards the increase of certified organic potato production. While several commercial EO products for potato sprout suppression already exist on the market, recent research is elucidating both the strengths and weaknesses of these various products as well as identifying other EOs with both sprout suppressing and enhancing properties. Future research in this field is necessary to fine-tune application rates and methods while also matching various EOs to the appropriate cultivars. However, there is great potential for not only longer storage capabilities, but also higher yields by using EOs in organic potato production and storage. Essential oils, generally regarded as safe, offer an enticing alternative to currently used growth regulators and pesticides such as CIPC for sprout

control. Such alternatives would reduce worker exposure to toxic chemicals, eliminate toxic residues in potato, and hence may significantly contribute to improved consumer health.

## 6. Methodology

To conduct the literature review, reputed search engines including Scopus, Pubmed, and Google Scholar databases were used with key words and phrases including "potato", "sprout suppression", "sprout inhibition", "storage", and "essential oil". The reference lists of included studies were hand-searched.

In this review, we included studies that assessed the effect of EOs or main components of commercially available EO products on sprout suppression in potato storage. This included studies that evaluated the impacts of temperature, suppressant concentration or dosage, or application frequency. Included studies were those that reported at least one effective EO or EO fraction. The review was limited to the English language.

**Author Contributions:** J.T.: Conceptualization, Writing—Original Draft, Writing—Review & Editing, Visualization. V.D.Z.: Conceptualization, Resources, Writing—Review & Editing, Supervision. All authors have read and agreed to the published version of the manuscript.

**Funding:** This work was partially supported by the Oregon Potato Commission, the Oregon Department of Agriculture—Specialty Crops BGP ODA6024GR, and by USDA-NIFA 2021-51106-35584 grants awarded to Valtcho Zheljazkov.

**Institutional Review Board Statement:** Not applicable.

**Informed Consent Statement:** Not applicable.

**Data Availability Statement:** Not applicable.

**Conflicts of Interest:** The funders had no role in the design of the study; in the collection, analyses, or interpretation of data; in the writing of the manuscript, or in the decision to publish the results.

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
