# Peer review of "Sprout Suppressants in Potato Storage: Conventional Options and Promising Essential Oils—A Review"

_sustainability, doi:10.3390/su14116382_

Round 1

Reviewer 1 Report

Dear authors,

The review paper ‘Achieving sprout suppression in stored potato: promising essential oils for fresh, processing, and seed potatoes’ is an attempt by authors to describe the use of the essential oil to suppress the sprouting in potatoes.

The subject is interesting and needs to be explored in depth.

Here are the comments to improve the overall manuscript.

  • Title: The title of the manuscript is misleading. The authors have not only discussed the role of essential oils for sprout suppression but also included an extensive review of the chemicals used for sprout suppression as well as the role of essential oils in disease suppression. The suggestion is to modify the title to a general one rather than a specific title like the one presented in the draft.
  • The manuscript has several typos, such as line 25: Typo ‘whence’; Line 133: Typo ‘4oC’. The manuscript needs proofreading.
  • Line 94. All the headings and subheadings should not have abbreviations.
  • Section 4. Essential Oils (EOs) for Pest and Disease Control in Stored Potatoes – This section is not required. Also, it lacks information. If the authors still want to keep this section, they need to describe this section.
  • Authors can also provide a life cycle of potato sprouting showing the timeline. Harvesting, processing, storing, and sprouting. This will add value to understanding which stages and after how many days sprouting starts and when is the treatment done.

Author Response

Dear authors,

The review paper ‘Achieving sprout suppression in stored potato: promising essential oils for fresh, processing, and seed potatoes’ is an attempt by authors to describe the use of the essential oil to suppress the sprouting in potatoes.

The subject is interesting and needs to be explored in depth.

Here are the comments to improve the overall manuscript.

  • Title: The title of the manuscript is misleading. The authors have not only discussed the role of essential oils for sprout suppression but also included an extensive review of the chemicals used for sprout suppression as well as the role of essential oils in disease suppression. The suggestion is to modify the title to a general one rather than a specific title like the one presented in the draft.

Response: The title has been modified to, ‘Sprout suppressants in potato storage: conventional options and promising essential oils. A review.’

  • The manuscript has several typos, such as line 25: Typo ‘whence’; Line 133: Typo ‘4oC’. The manuscript needs proofreading.

Response: Line 25: ‘whence’ has been changed to ‘from which’. Line 206: ‘4oC’ has been changed to ‘4°C’. The manuscript has been reviewed again and other typos were corrected.

  • Line 94. All the headings and subheadings should not have abbreviations.

Response: Corrected as suggested

  • Section 4. Essential Oils (EOs) for Pest and Disease Control in Stored Potatoes – This section is not required. Also, it lacks information. If the authors still want to keep this section, they need to describe this section.

Response: Section 4 Essential Oils for Pest and Disease Control in Stored Potatoes has been removed.

  • Authors can also provide a life cycle of potato sprouting showing the timeline. Harvesting, processing, storing, and sprouting. This will add value to understanding which stages and after how many days sprouting starts and when is the treatment done.

Response: A paragraph has been added discussing pre-storage and timing of sprout suppression methods (Lines 39-56). It reads,

”Once harvested, potatoes may first undergo a "pre-storage" phase where they are stored at 95% humidity between 10 and 15°C for two weeks [2]. This phase allows potatoes to heal their peels after potentially being damaged during harvest while also allowing the potatoes to dry. Following pre-storage, potatoes are stored at low temperatures in piles or crates for periods ranging from several weeks to many months during which additional chemical or physical methods of sprout suppression may be applied prior to dormancy break [7].”

Reviewer 2 Report

Detailed comments for Authors
1) Line 213 - Latin name, italics.
2) Line 67 - Consider supplementing the physical methods with electromagnetic radiation (EF). Research on the influence of the magnetic field has shown that it (MF) can act as both an inhibitor and activator of sprouting. The situation is similar in the case of microwaves (especially 2.45 GHz). Researchers indicate the effect of specific doses of radiation (exposure time, etc.). Note that ultraviolet (especially UV-C) was also used in the storage processes, 250–280 nm has the ability to damage plant cells (253.7 nm damages DNA). Currently, research is underway on the possibility of using neutrons in plant stimulation (unpublished research).
3) The drawback of the work is the authors' focus on chemical issues, the physical interactions are briefly described. I understand the authors that many physical methods are not (and will not be) approved for the food industry (food and consumer safety). However, the work is of a review nature and all aspects influencing the sprouting of potato tubers should be taken into account (at least in the "Introduction").
4) It should be clearly emphasized in the manuscript that the factor determining (regulating) the germination process (during storage of potato tubers) is the temperature and humidity inside the cold store. Chemical (based on natural factors) and physical methods are only an addition.
5) Publications that comprehensively deal with the impact of physical methods on stored potato tubers: DOI10.3390 / su12031048, EFFECTS OF MICROWAVE RADIATION ON THE GERMINATION OF SOLANUM TUBEROSUM L. TUBERS, DOI10.3329 / bjb.v47i3.38722, DOI10.15199 / 48.2020.01.36

Author Response

Detailed comments for Authors
1) Line 213 - Latin name, italics.

Response: Corrected as suggested

2) Line 67 - Consider supplementing the physical methods with electromagnetic radiation (EF). Research on the influence of the magnetic field has shown that it (MF) can act as both an inhibitor and activator of sprouting. The situation is similar in the case of microwaves (especially 2.45 GHz). Researchers indicate the effect of specific doses of radiation (exposure time, etc.). Note that ultraviolet (especially UV-C) was also used in the storage processes, 250–280 nm has the ability to damage plant cells (253.7 nm damages DNA). Currently, research is underway on the possibility of using neutrons in plant stimulation (unpublished research).

Response: Several sources have been added to the section discussing physical methods of sprout suppression including those concerning microwave, gamma, and ultraviolet irradiation, and pressure and magnetic treatments (Lines 87-90).

3) The drawback of the work is the authors' focus on chemical issues, the physical interactions are briefly described. I understand the authors that many physical methods are not (and will not be) approved for the food industry (food and consumer safety). However, the work is of a review nature and all aspects influencing the sprouting of potato tubers should be taken into account (at least in the "Introduction").

Response: Additional physical methods of sprout suppression have been included in the introduction, although lengthy discussion of these methods is avoided as the main focus of the review is on chemical methods (Lines 87-90).

4) It should be clearly emphasized in the manuscript that the factor determining (regulating) the germination process (during storage of potato tubers) is the temperature and humidity inside the cold store. Chemical (based on natural factors) and physical methods are only an addition.

Response: The importance of temperature and humidity level on dormancy break is now clearly stated (Lines 78-84).

5) Publications that comprehensively deal with the impact of physical methods on stored potato tubers: DOI10.3390 / su12031048, EFFECTS OF MICROWAVE RADIATION ON THE GERMINATION OF SOLANUM TUBEROSUM L. TUBERS, DOI10.3329 / bjb.v47i3.38722, DOI10.15199 / 48.2020.01.36

Response: Thank you for the additional literature. Several references to studies concerning physical methods of sprout suppression have been added to the review, including:

10.17660/ActaHortic.2018.1194.69

10.1007/s13197-011-0337-9

10.1111/j.1365-2621.2010.02455.x

10.22616/ERDev2019.18.N307

Jakubowski, Tomasz. (2016). Effect of microwave radiation on the germination of Solanum tuberosum L. tubers. 45. 1253-1255.

Reviewer 3 Report

There are no reservations to the subject of the manuscript, which is suitable for the Sustainability Journal. The review presents an analysis of various methods of potato sprout suppression, in particular with the application of ethereal oils  used to suppress potato sprouting. The subject of the review presented by the authors is very significant and can be used in agricultural practice, because potato sprouting during storage is a serious problem, which leads to loss of income and food waste.  

There are no major reservations to the abstract. It contains information about the content of the manuscript.

Introduction:

The content of the introduction is appropriate. The authors clearly specified the aim of the considerations. The Discussion is well structured, the reviewed research results are confronted with other authors’ research. However, the manuscript contain several instances of incorrect referencing:

Line 210 – is: Rylski et al. (1974) – please adjust to the journal referencing style.

Line 213 – is: “Mentha spicata”, please italicise

Line 265 – is: Emragi et al. (2022) – please adjust to the journal referencing style.

Line 294 – is: Horper (2019), like in line 302 and line 314 – please adjust to the journal referencing style.

Line 322 – is: Hortmans et al. (1995) – please adjust to the journal referencing style.

Line 368 – is: Visse-Mansiaux et al. (2021) – please adjust to the journal referencing style.

Line 370– is: Koradogan (2019) – please adjust to the journal referencing style.

Line 388 – is: Finger et al. (2018) – please adjust to the journal referencing style.

Line 406 – is: Frazier et al. (2004) – please adjust to the journal referencing style.

Line 414 – is: Finger et al. (2018) – please adjust to the journal referencing style.

Line 420 – is: Song et al. (2009) – please adjust to the journal referencing style.

Line 475 – is: Owolabi et al. (2010, 2013) – please adjust to the journal referencing style.

Line 493 – is: Belay et al. (2022) – please adjust to the journal referencing style.

Line 494 – is: Li et al. (2022) – please adjust to the journal referencing style.

Line 536 – is: Goodarzi et al. (2016) – please adjust to the journal referencing style

Line 548 – is: Shukla et al. (2019) – please adjust to the journal referencing style.

Line 559 – is: Song et al. (2007) – please adjust to the journal referencing style.

Line 579 – is: Song et al. (2009) – please adjust to the journal referencing style.

References

All references fail to comply with the referencing style guide of the Sustainability Journal – please correct them.

Subject to the adjustments, the paper can be published in the Sustainability Journal.

Author Response

There are no reservations to the subject of the manuscript, which is suitable for the Sustainability Journal. The review presents an analysis of various methods of potato sprout suppression, in particular with the application of ethereal oils  used to suppress potato sprouting. The subject of the review presented by the authors is very significant and can be used in agricultural practice, because potato sprouting during storage is a serious problem, which leads to loss of income and food waste.  

Response: Thank you.

There are no major reservations to the abstract. It contains information about the content of the manuscript.

Response: Thank you.

Introduction:

The content of the introduction is appropriate. The authors clearly specified the aim of the considerations. The Discussion is well structured, the reviewed research results are confronted with other authors’ research. However, the manuscript contain several instances of incorrect referencing:

Line 210 – is: Rylski et al. (1974) – please adjust to the journal referencing style.

Response: Corrected as suggested – see highlighting for all adjusted references in the text.

Line 213 – is: “Mentha spicata”, please italicize

Response: Corrected as suggested

Line 265 – is: Emragi et al. (2022) – please adjust to the journal referencing style.

Response: Corrected as suggested

Line 294 – is: Horper (2019), like in line 302 and line 314 – please adjust to the journal referencing style.

Response: Corrected as suggested

Line 322 – is: Hortmans et al. (1995) – please adjust to the journal referencing style.

Response: Corrected as suggested

Line 368 – is: Visse-Mansiaux et al. (2021) – please adjust to the journal referencing style.

Response: Corrected as suggested

Line 370– is: Koradogan (2019) – please adjust to the journal referencing style.

Response: Corrected as suggested

Line 388 – is: Finger et al. (2018) – please adjust to the journal referencing style.

Response: Corrected as suggested

Line 406 – is: Frazier et al. (2004) – please adjust to the journal referencing style.

Response: Corrected as suggested

Line 414 – is: Finger et al. (2018) – please adjust to the journal referencing style.

Response: Corrected as suggested

Line 420 – is: Song et al. (2009) – please adjust to the journal referencing style.

Response: Corrected as suggested

Line 475 – is: Owolabi et al. (2010, 2013) – please adjust to the journal referencing style.

Response: Corrected as suggested

Line 493 – is: Belay et al. (2022) – please adjust to the journal referencing style.

Response: Corrected as suggested

Line 494 – is: Li et al. (2022) – please adjust to the journal referencing style.

Response: Corrected as suggested

Line 536 – is: Goodarzi et al. (2016) – please adjust to the journal referencing style

Response: Corrected as suggested

Line 548 – is: Shukla et al. (2019) – please adjust to the journal referencing style.

Response: Corrected as suggested

Line 559 – is: Song et al. (2007) – please adjust to the journal referencing style.

Response: Corrected as suggested

Line 579 – is: Song et al. (2009) – please adjust to the journal referencing style.

Response: This section has been removed.

References

All references fail to comply with the referencing style guide of the Sustainability Journal – please correct them.

Response: All references have been reformatted to follow MDPI style guidelines.

Subject to the adjustments, the paper can be published in the Sustainability Journal.

Reviewer 4 Report

This study reviewed various methods of sprout suppression in potatoes.  At the turn of 2019/2020, the European Commission decided to ban the use of chlorpropham, as well as several other chemical substances popular in agriculture, so the topic is timely.  The search for new products based on essential oils fits well with the journal's range related to sustainable development in agriculture.

Overall, the article is well written and provides an interesting overview of essential oil uses for fresh potatoes, potatoes for further processing, seed potatoes and disease control in stored potatoes. However the clarity of the overall presentation is not satisfactory. Table should include more information such as: examined cultivars and main advantages and disadvantages applied essential oils. In summary, it would be worth pointing to examples of these essential oils that are the best alternative to withdrawn preparations.

Other comments

Line 34 “metric ton” –It is better to apply International System of Units

Table 1. It would be worth rebuilding the table structure. In its present form, it takes up a lot of space, and at the same time many fields are empty.

Author Response

This study reviewed various methods of sprout suppression in potatoes.  At the turn of 2019/2020, the European Commission decided to ban the use of chlorpropham, as well as several other chemical substances popular in agriculture, so the topic is timely.  The search for new products based on essential oils fits well with the journal's range related to sustainable development in agriculture.

Response: Thank you.

Overall, the article is well written and provides an interesting overview of essential oil uses for fresh potatoes, potatoes for further processing, seed potatoes and disease control in stored potatoes. However the clarity of the overall presentation is not satisfactory. Table should include more information such as: examined cultivars and main advantages and disadvantages applied essential oils. In summary, it would be worth pointing to examples of these essential oils that are the best alternative to withdrawn preparations.

Response: Thank you. The table has been updated to include the cultivars tested in each study. Comparisons of essential oil applications to synthetic chemicals such as CIPC are made whenever possible and within the scope of the included studies. As essential oil application on potato sprouting can vary greatly with cultivar, storage temperature, and application scheme, it is difficult to make broad recommendations about which treatments make the best alternatives to withdrawn preparations. Rather, recommendations may be made on a case-by-case basis, but we believe this is beyond the scope of this review.

Other comments

Line 34 “metric ton” –It is better to apply International System of Units

Response: Corrected as suggested (Lines 34-35)

Table 1. It would be worth rebuilding the table structure. In its present form, it takes up a lot of space, and at the same time many fields are empty.

Response: The columns corresponding to “Impacts on Tuber Yield” and “Impacts on Flavor” have been removed to address the many empty fields in the table. Table column and row widths have been adjusted to minimize the table area.